# Three-dimensional solitary waves with electrically tunable direction of propagation in nematics

Bing-Xiang Li [1,2], Rui-Lin Xiao[1,2], Sathyanarayana Paladugu[1], Sergij V. Shiyanovskii [1,2] & Oleg D. Lavrentovich [1,2,3]

Production of stable multidimensional solitary waves is a grand challenge in modern science. Steering their propagation is an even harder problem. Here we demonstrate three-dimensional solitary waves in a nematic, trajectories of which can be steered by the electric field in a plane perpendicular to the field. The steering does not modify the properties of the background that remains uniform. These localized waves, called director bullets, are topologically unprotected multidimensional solitons of $(3 + 2)$D type that show fore-aft and right-left asymmetry with respect to the background molecular director; the symmetry is controlled by the field. Besides adding a whole dimension to the propagation direction and enabling controlled steering, the solitons can lead to applications such as targeted delivery of information and micro-cargo.

[1] Advanced Materials and Liquid Crystal Institute, Kent State University, Kent, OH 44242, USA. [2] Chemical Physics Interdisciplinary Program, Kent State University, Kent, OH 44242, USA. [3] Department of Physics, Kent State University, Kent, OH 44242, USA. Correspondence and requests for materials should be addressed to O.D.L. (email: olavrent@kent.edu)

Solitary waves maintain their self-confined shapes while propagating and surviving collisions with each other. A classic example is a one-dimensional wave in a shallow water channel observed by Russell[1,2]. Creation of solitons of higher dimensions represents a major challenge in the science of nonlinear fields and matter[3]. The multidimensional solitons are abbreviated as $(m + 1)$D objects, where "1" represents the propagation direction and "$m$" shows in how many dimensions the soliton is self-focused[4,5]. The Russell's wave is a $(1 + 1)$D type, as it propagates along the channel and water displacement occurs along a single transverse direction. Multidimensional solitons are of practical importance[3], for example, in optics, where the so-called "light bullets" of a $(3 + 1)$D type could serve as information carriers[6]. Among the broad family of solitons there is a class of localized structures, often called dissipative solitons[7–11]. Dissipative solitons require an external driving; they represent a portion of a pattern surrounded by a homogeneous steady state; below a certain strength of the driver, they vanish[7–11]. Experimentally, dissipative solitons were realized in the form of electric current filaments in a 2D planar gas-discharge system[12] and also as $(3 + 1)$D particle-like electrically powered solitary waves of molecular orientation (specified by the so-called director $\hat{\mathbf{n}}$) in a nematic liquid crystal[13]. These $(3 + 1)$D solitons, called "director bullets", represent spatially-confined perturbations of the director field $\hat{\mathbf{n}}(\mathbf{r})$ that coexist with a uniform director state $\hat{\mathbf{n}}_0 = \text{const}$. The distortions $\hat{\mathbf{n}}(\mathbf{r})$ oscillate with the same frequency as the frequency $f$ of the driving electric field. They disappear when the amplitude of the electric field becomes too low and when $f < 100$ Hz.

A grand challenge of soliton research is whether and how direction of soliton propagation could be guided[5]. In a uniform medium, solitons propagate along rectilinear trajectories or experience Brownian motion, see, for example, refs. [12,14] describing dissipative solitons. In particular, director bullets of $(3 + 1)$D type lack fore-aft symmetry and move perpendicularly to the electric field and the background director $\hat{\mathbf{n}}_0$, along rectilinear trajectories[13]; the polarity of propagation is determined by the polarity of bend deformation of the director which breaks the fore-aft symmetry[13]. A similar case of localized structures moving in one spatial dimension as a result of an internal symmetry breaking instability has been described independently and simultaneously by Alvarez-Socorro, Clerc and Tlidi[14] for a 2D isotropic system. In contrast to the case of $(3 + 1)$D director bullets, the motion direction of localized structures considered in ref. [14] is arbitrary (defined by the initial condition used in simulations): a preferred direction does not exist since the system is isotropic[14]. To control trajectories of solitons and to produce trajectories different from rectilinear or Brownian-type, one could use interactions of solitons with other waves[4], with each other[15], or by designing a spatially varying medium. One example are nematicons, self-focused light beams propagating in a nematic:[3,5] these can be bent by a spatially varying director $\hat{\mathbf{n}}(\mathbf{r})$[16].

In this work, we demonstrate experimentally multidimensional steerable solitary waves of a $(3 + 2)$D director bullets type. These director bullets represent waves of director deformations driven by an alternating current (AC) electric field. The solitons form in a planar homogeneous sandwich-like nematic cell, with the ground-state director $\hat{\mathbf{n}}_0$ being parallel to the bounding plates and to planar transparent electrodes. The electric field, applied along the normal to the cell, is perpendicular to $\hat{\mathbf{n}}_0$. The waves are self-localized in three spatial dimensions, while the background $\hat{\mathbf{n}}_0$ remains unperturbed. The dynamic director structure of the solitons is established by polarizing optical microscopy in transmission mode with ultrafast video camera. The patterns of transmitted light intensity for normal and oblique incidence are analyzed by the theory based on Berreman's $4 \times 4$ matrix formalism[17]. The analysis establishes that the director deformations within the solitons oscillate with the frequency of the driving electric field. This feature points to the linear coupling of $\hat{\mathbf{n}}(\mathbf{r})$ to the electric field as a main reason for the bullets stability. Most interestingly, depending on the electric field amplitude, the solitons show different geometries of symmetry breaking of the director deformations $\hat{\mathbf{n}}(\mathbf{r})$ and because of that, they can move either parallel to $\hat{\mathbf{n}}_0$ or perpendicular to it. The electric field, being perpendicular to the bullets trajectories, provides no sense of direction by itself; it is the in-plane anisotropy of the system and its coupling to the field that produces different scenarios of symmetry breaking and allows one to steer the solitons by changing the electric field. To stress the 2D control over trajectories of the director bullets, we classify them as $(3 + 2)$D solitons. To the best of our knowledge, there are no other examples of stable and steerable multidimensional solitons that move through a homogeneous background along externally controlled trajectories, neither among the "classic" solitons in conservative systems (such as the Russell's solitary wave) nor among the localized structures/dissipative solitons in driven dissipative systems.

## Results

**Materials and experimental design.** We used a nematic 4′-butyl-4-heptyl-bicyclohexyl-4-carbonitrile (CCN-47) doped with 0.005 wt% of ionic salt tetrabutylammonium bromide (TBABr). The material is of the $(-, -)$ type, with negative dielectric $\Delta\varepsilon = -3.3$ and conductivity $\Delta\sigma = -2 \times 10^{-9}\ \Omega^{-1}\ \text{m}^{-1}$ anisotropy. The cell of thickness $d = 8.0\ \mu\text{m}$ is composed of two glass substrates coated with transparent electrodes and surface alignment layers to set planar orientation $\hat{\mathbf{n}}_0 = (0, 1, 0)$ in the $xy$ plane parallel to the bounding plates. A sinusoidal AC electric field of frequency $f = 1–10^3$ Hz is normal to the substrates, $E = (0, 0, E)$.

**Diagram of states and soliton stability.** In the range 5 Hz $< f <$ 27 Hz, the electric field produces a new type of solitary waves of director deformations, which are $(3 + 2)$D director bullets abbreviated as $B_\alpha^l$; the superscript means "low frequency" and the subscript indicates the angle $\alpha$ between $v$ and $\hat{\mathbf{n}}_0$. At $U = \text{const}$, $B_\alpha^l$ solitons move either parallel to $\hat{\mathbf{n}}_0$, Fig. 1, Supplementary Movie 1, or perpendicularly to it, Fig. 2, Supplementary Fig. 1, Supplementary Movie 2. We call these $B_0^l$ and $B_{90}^l$ bullets, respectively. Their length and width of the bullets in the $xy$ plane is in the range 15–30 μm, which is many orders of magnitude smaller than the extension of the nematic cell (5 mm × 5 mm). Solitons appear at imperfections such as dust particles, Fig. 3, electrode edges, but also might appear from locations in which the presence of impurities is hard to detect.

The stability range of $B_\alpha^l$ is limited by frequency, 5 Hz $\leq f \leq$ 27 Hz, and voltage, 4 V $\leq U \leq$ 13 V, Fig. 4a. At very high frequencies, $f \geq 500$ Hz, one also observes director bullets of the $(3 + 1)$D type, Fig. 4b, abbreviated $B_{90}^h$ (the superscript refers to "high frequency") and described previously[13]. These $(3 + 1)$D bullets move only perpendicularly to $\hat{\mathbf{n}}_0$ and would not be discussed any further. The experimentally established range underlines that the $B_\alpha^l$ stability results from a fine balance of different mechanisms of the electric field-nematic coupling, such as flexoelectric polarization, anisotropy of dielectric permittivity, conductivity and its anisotropy. Absence of solitons at very low frequencies $f < 5$ Hz is apparently caused by screening of electrodes by ions.

At $f = \text{const}$, the voltage increase produces first $B_0^l$ solitons, then coexisting $B_0^l$ and $B_{90}^l$, then solely $B_{90}^l$, Fig. 4c; the speed of solitons grows with $U$, Fig. 4d. At $U > U_{\text{EHD}}$ $(f) \approx (12.8 + 0.08\text{s}\ f)$ V, one observes periodic director domains, Fig. 4a, that cover the entire electrode area of the cells, Fig. 5.

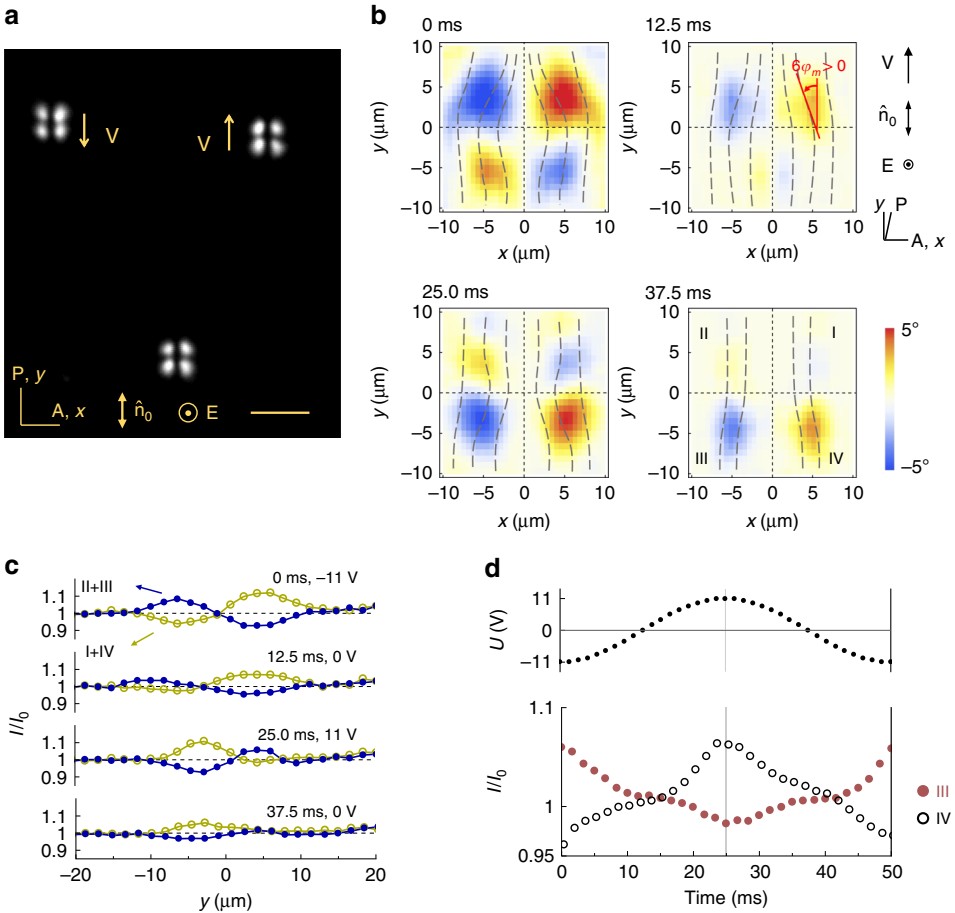

**Fig. 1** $B_0^l$ solitons driven by voltage 8.0 V and frequency 20 Hz. **a** Polarizing microscope texture. Scale bar 30 μm. **b** Azimuthal director distortions $\varphi_m(x, y, t)$ reproduced from polarizing microscopy with two polarizers crossed at 78°. The time step between images is 1/4 of the voltage period, with "0 ms" corresponding to the negative extremum of the voltage; $\sqrt{2}\,U \approx 11.3$ V. **c** y-profile of the transmitted light intensity integrated over the x-axis in right (I + IV) and left (II + III) parts of the bullet at different voltages/times. **d** Time/voltage dependence of light intensity transmitted through quadrants III and IV. In **c**, **d**, $I$ is normalized by the transmitted intensity $I_0$ outside the bullet

Deterministic directional propagation of $B_\alpha^l$ bullets differs dramatically from the stochastic motion of localized structures of current filaments described experimentally[12] and theoretically[14]. The obvious reason is the anisotropy of the nematic cell. Similarly to the (3 + 1)D $B_{90}^h$ solitons described previously[13], propagation direction of the $B_\alpha^l$ bullets is associated with the internal symmetry breaking. However, the important new feature of $B_\alpha^l$ is that this internal symmetry breaking can be either fore-aft ($B_0^l$, Fig. 1) or left-right ($B_{90}^l$, Fig. 2), or producing no mirror symmetry at all, Fig. 6.

**Electric field control of soliton symmetry and trajectories**. The most striking feature of the $B_\alpha^l$ bullets is that their symmetry and thus trajectories are controlled by the electric field. A step-like increase of $U$ from 8.0 to 11.0 V at $f =$ const, transforms $B_0^l$ into $B_{90}^l$, Fig. 6a, while a decrease of $U$ transforms $B_{90}^l$ into $B_0^l$, Fig. 6b and Supplementary Movies 3 and 4. In both cases, the bullets change their propagation direction by 90°, Fig. 6c; the trajectories can also be controlled by the voltage change rates, Fig. 6d, e.

The solitons exhibit an intriguing behavior near the limits of the stability islands. Once the voltage is increased above ≈12.5 V (at 20 Hz), the $B_{90}^l$ solitons stop. If the voltage remains fixed, they disappear within the decay time $\tau \approx 2$–3 s that depends on the depth of voltage increase. However, if the voltage is reduced back within $\tau$, the solitons start to move, either in the same direction or

in the opposite direction, Fig. 7a, b, Supplementary Movie 5. The outcome depends on the exact timing of voltage change. Similar effects of stoppage, disappearance, and reactivation are observed for $B_0^l$, when the voltage is reduced below 7.2 V (at 20 Hz), Fig. 7c, d, and Supplementary Movie 6. Note an important symmetry effect: when the bullets stop, the director field becomes left-right and fore-aft symmetric; a resumed motion coincides with restoration of the asymmetric shapes, Fig.7a, c.

**Interactions of solitons**. The $B_\alpha^l$ bullets reveal their soliton nature by surviving collisions, Fig. 8. Within different stability islands, solitons show different outcomes of collisions. If only one soliton type is stable, their collision results in reappearance of a similar pair. In Fig. 8a, b, two $B_0^l$ bullets move parallel to $\hat{\mathbf{n}}_0$ towards each other, coalesce and form a single perturbation that separates into two $B_{90}^l$ solitons that move perpendicularly to $\hat{\mathbf{n}}_0$, Supplementary Movie 7. This symmetry change is only temporary, as the two reconstruct their $B_0^l$ structure and continue propagation parallel to $\hat{\mathbf{n}}_0$. When collisions happen in the phase diagram region where both types of solitons are stable, new channels of reaction emerge. For example, two $B_0^l$ solitons can merge into a single $B_{90}^l$ that propagates perpendicularly to the "parent" solitons, Fig. 8c, d, Supplementary Movie 8. An opposite effect is also possible, with two $B_{90}^l$'s producing a single $B_0^l$. Collisions of $B_0^l$ and $B_{90}^l$ produce other scenarios, such as transformations $B_0^l \rightarrow B_{90}^l$ and $B_{90}^l \rightarrow B_0^l$.

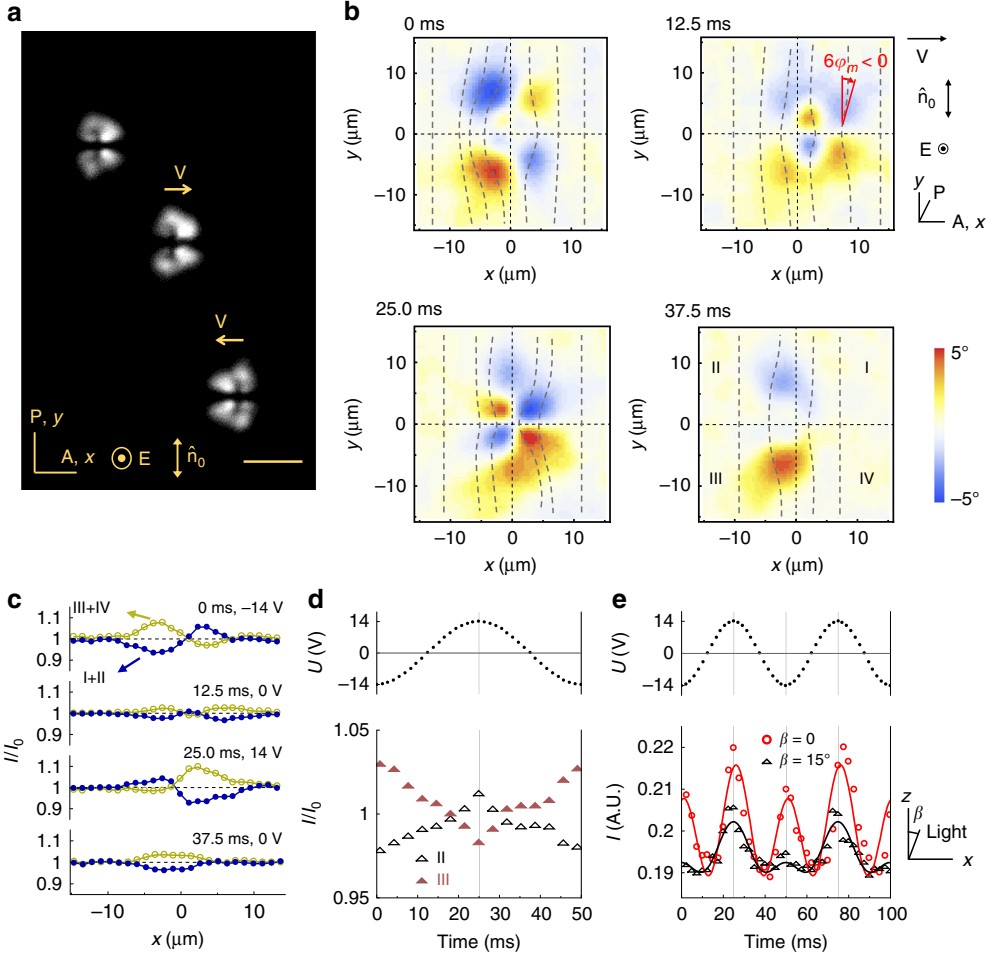

**Fig. 2** $B_{90}^l$ solitons driven by voltage 10.0 V and 20 Hz. **a** Polarizing microscope texture. Scale bar 30 μm. **b** Azimuthal director distortions $\varphi_m(x, y, t)$ reproduced with two polarizers crossed at 65°. The time step between images is 1/4 of the voltage period, time "0" corresponds to the negative extremum of the voltage; $\sqrt{2}\, U \approx 14.1$ V. **c** x-profile of the light intensity integrated over the y-axis in top (I + II) and bottom (III + IV) parts of the bullet at different voltages/times. **d** Time/voltage dependence of light intensity transmitted through quadrants II and III. **e** Dynamics of light intensity at the quadrant I for the crossed polarizers at normal, $\beta = 0$, and oblique incidence, $\beta = 15°$. The solid lines show the fitting with Eq. (1) derived in Methods. In **c**, **d**, I is normalized by the transmitted intensity $I_0$ outside the bullet

**In-plane director structure of the solitons**. To understand the unusual properties of the (3 + 2)D solitons, we explore their inner structure. The direction of propagation of the bullets is determined by the dynamic asymmetry of the in-plane director rotations $\varphi(x, y, z, t)$ away from $\hat{\mathbf{n}}_0$. Here t is time. Because of surface anchoring, $\varphi = 0$ at the bounding plates, and reaches its maximum in the middle plane of the cell, $\varphi(x, y, z = d/2, t) = \varphi_m(x, y, t)$. In observations with two crossed polarizers, a nonzero $\varphi_m$ increases light transmittance through the sample. The polarizing optical microscope textures of $B_0^l$ and $B_{90}^l$ represent four quadrants of a nonzero $\varphi_m$, surrounded by a uniform ($\varphi = 0$) background, Figs. 1a and 2a, Supplementary Movies 1 and 2. The $B_0^l$ bullets propagating along $\hat{\mathbf{n}}_0$ lack the fore-aft symmetry, Fig. 1a. The $B_{90}^l$ bullets, propagating perpendicularly to $\hat{\mathbf{n}}_0$, lack the left-right symmetry, Fig. 2a. This intrinsic symmetry breaking of the director field is reminiscent of the symmetry breaking that leads to a transition from the stationary to moving localized structures/dissipative solitons described by Alvarez-Socorro et al.[14] with that difference that the nematic background is anisotropic and thus the spectrum of possible symmetry breaking scenarios for director bullets is much broader, ranging from left-right to fore-aft mirror symmetries and to structures that have no mirror symmetry at all.

To map the spatiotemporal variations of $\varphi_m$, we use two linear polarizers crossed at an angle of 78° or 65° with each other. Usually, polarizing microscopy is performed with the two polarizers crossed at 90°. However, such standard approach does not allow one to distinguish the states $\varphi$ and $-\varphi$. The decrossed polarizers allow one to distinguish $\varphi$ and $-\varphi$; the angular values of decrossing, 78° or 65°, have been found experimentally to yield a good contrast between the orientations $\varphi$ and $-\varphi$. The light intensity I transmitted through the soliton changes in time with the same frequency as the frequency f of the AC field, Figs. 1b–d and 2b–d. The intensity I depends on $\varphi_m$. In Figs. 1b and 2b, counterclockwise rotation $\varphi_m > 0$ results in a higher I, while $\varphi_m <$ 0 reduces I. By using Jones matrices[18], we calculate $I(x, y, t)$ as a function of $\varphi_m(x, y, t)$ and plot the director for a single period of the AC field in Figs. 1b and 2b. The azimuthal reorientations are weak, reaching a maximum $\varphi_{max} \approx 5°$, for both $B_0^l$ and $B_{90}^l$. For clarity, the tilts of the director in Figs. 1b and 2b are enlarged by a factor of 6.

**Out-of-plane director structure of the solitons**. The director also experiences oscillations of the polar angle $\theta(x, y, z, t)$ measured with respect to the $xy$ plane, with the same frequency f, Fig. 2e. To prove these polar oscillations, we measured the

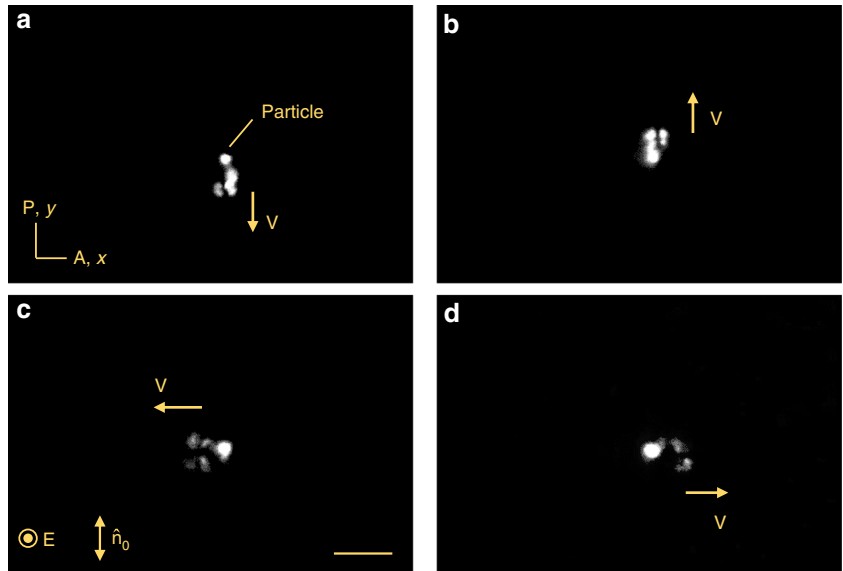

**Fig. 3** Nucleation of $B_0^l$ and $B_{90}^l$ solitons at the same dust particle. The solitons $B_0^l$ nucleate and move **a** down and **b** up, parallel to the background director. The solitons $B_{90}^l$ nucleate and move **c** left and **d** right, perpendicularly to the background director. Temperature 55 °C; applied voltage 9.0 V, frequency 20 Hz, scale bar 30 μm

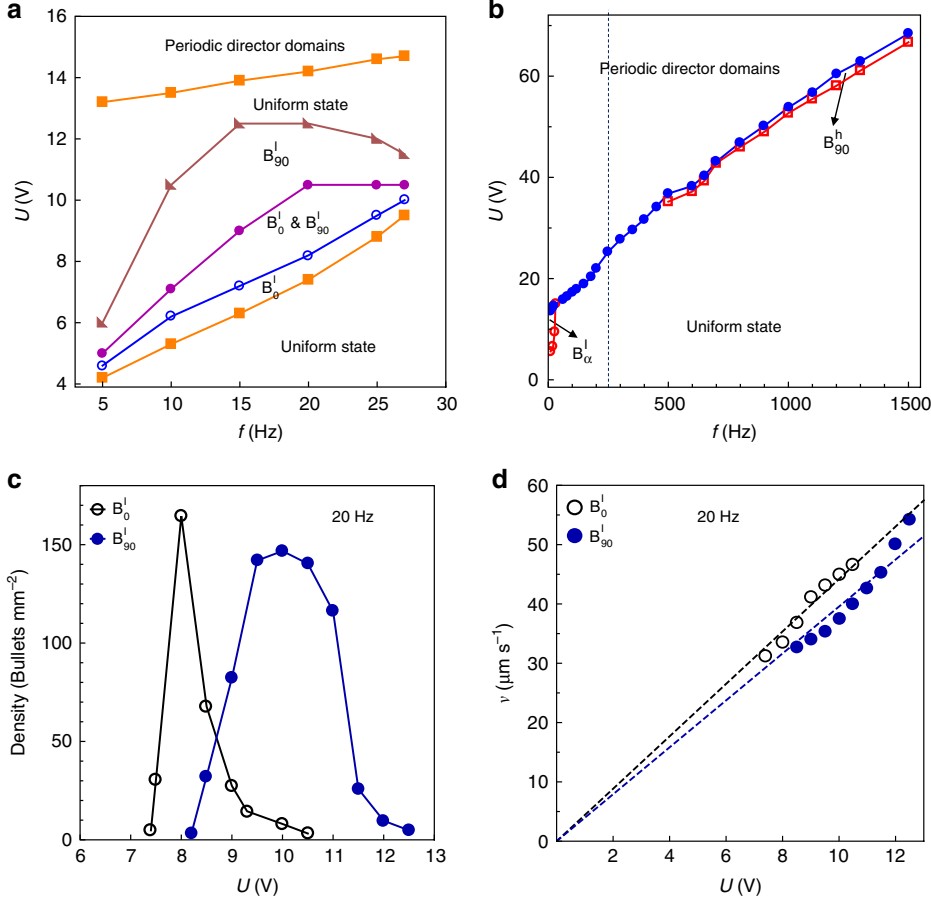

**Fig. 4** Soliton properties as a function of the applied voltage. **a** Phase diagram of director bullets $B_\alpha^l$ at low frequencies. **b** Phase diagram of director bullets $B_\alpha^l$ and $B_{90}^h$ in the broad range of frequencies explored for the same nematic cell. $B_{90}^h$ are (3 + 1)D director bullets. **c** Densities and **d** velocities of $B_0^l$ and $B_{90}^l$ as a function of the applied voltage. All data correspond to 55 °C

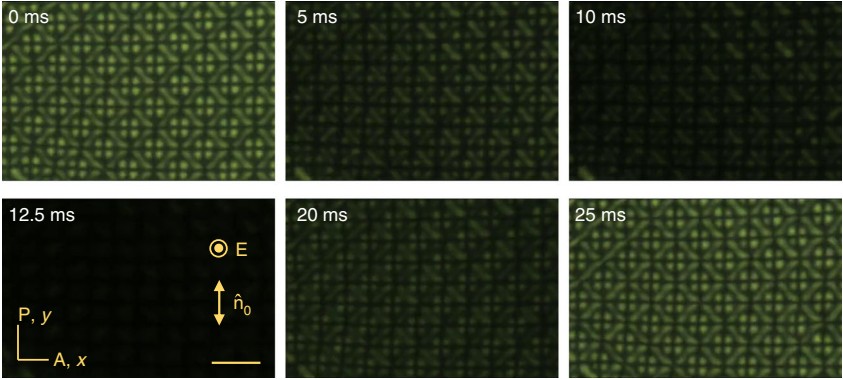

**Fig. 5** Polarizing microscopy textures of oscillating patterns. The patterns cover the entire electrode area of the cell and appear at voltages higher than the range of stability of isolated solitons. The images are taken at the moments of time indicated on the textures, within a 25 ms time interval that corresponds to the half period of the applied voltage (14.7 V, 20 Hz). Scale bar 20 μm

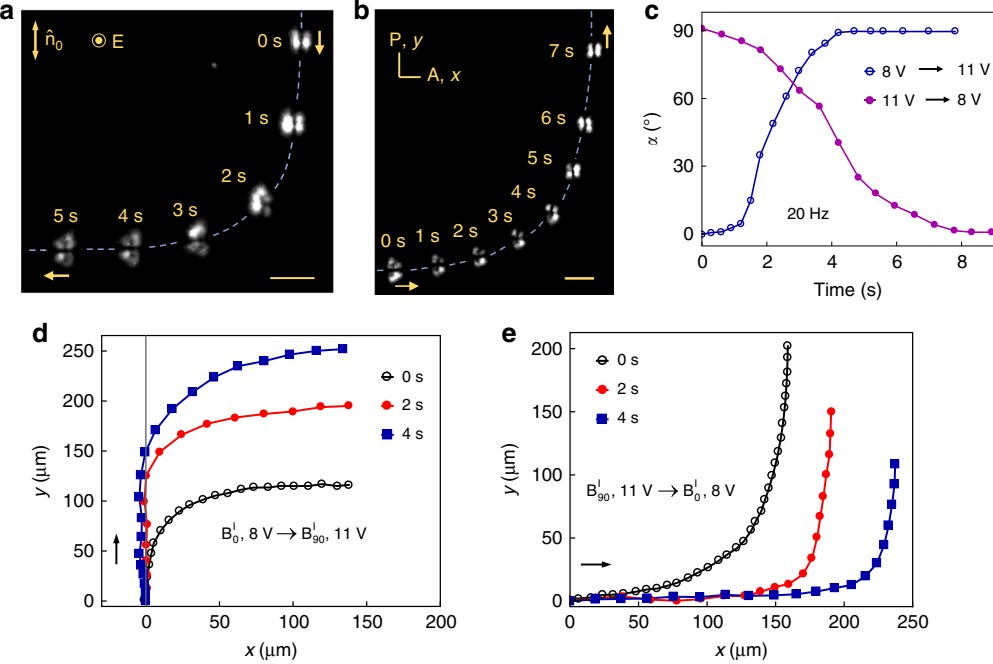

**Fig. 6** Electric field control of soliton symmetry and trajectories. **a** $B_0^l$ transforms into $B_{90}^l$ when the voltage is raised at $t = 0$ s from 8.0 to 11.0 V. Scale bar 30 μm. **b** The soliton $B_{90}^l$ transform into $B_0^l$ when the voltage at $t = 0$ s is decreased from 11.0 to 8.0 V. Scale bar 30 μm. **c** Time dependence of the angle $\alpha$ between the propagation direction and $\hat{\mathbf{n}}_0$ for transformations shown in **a** and **b**. **d** Changes of propagation directions during $B_0^l \rightarrow B_{90}^l$ and **e** $B_{90}^l \rightarrow B_0^l$ transformation for different rates of the voltage change from 8 to 11 V and back; the voltage is ramped linearly within 0 s, 2 s, and 4 s

dynamics of light transmittance through the cell and two crossed polarizers at oblique incidence of the probing light beam, $\beta = 15°$, Fig. 2e, where $\beta$ is the angle between the probing beam and the normal to the cell measured outside the cell. The theoretical model in "Material and methods" section, Eq. (11), predicts that for $B_{90}^l$ solitons, when the incidence plane $xz$ is perpendicular to $\hat{\mathbf{n}}_0$, the transmitted light intensity $I$, normalized by the incident intensity, depends on the angle of incidence $\beta$, the azimuthal $\varphi_m$ and polar $\theta_m$ director angles measured in the middle plane,

$$I \propto I_{\text{leak}} + \frac{4\Gamma^2}{\pi^2}\left(1 - 0.04\Gamma^2\right)\left(\varphi_m - \theta_m \tan \beta_{\text{LC}}\right)^2, \quad (1)$$

where $I_{\text{leak}}$ is the normalized light intensity leaked while the beam propagates between the crossed polarizers and the cell, $\Gamma = 2\pi\Delta nd/\lambda \approx 1.7$ is the phase retardation of the undistorted nematic, determined by the wavelength $\lambda = 530$ nm and birefringence $\Delta n = n_e - n_o$, $n_o$, and $n_e$ are the ordinary and extraordinary

refractive indices, respectively; $\beta_{\text{LC}} = \beta/n_o$ is the angle at which the probing beam propagates inside the nematic. For $\beta_{\text{LC}} = \beta = 0$, $I$ depends only on $\varphi_m$; however, for $\beta_{\text{LC}} = \beta \neq 0$, Eq. (1) demonstrates that the measured light intensity depends on the sign of $\theta_m$, which allows us to determine the frequency of its oscillations. We found experimentally that the value $\beta = 15°$(which corresponds to $\beta_{\text{LC}} = \beta/n_o = 10.6°$) provides the best contrast between $\theta_m$ and $-\theta_m$. The experiment, Fig. 2e, demonstrates that the transmission peaks for the two half-periods of the applied field are different for both normal $\beta_{\text{LC}} = \beta = 0$ and oblique $\beta = 15°$ light incidence. Since $I \propto \varphi_m^2$, Eq. (1), the asymmetry of the two half-periods indicates that angle $\varphi_m$ has a nonzero average value and changes with the frequency $f$. The experiment and modeling for the oblique $\beta = 15°$ light incidence, Fig. 2e, show that $\theta_m$ oscillates with the same frequency $f$. Figure 2e and Eq. (1) demonstrate that the variations of $\theta_m$ in the I quadrant are in phase with the variations of $\varphi_m$, as the term $\theta_m \tan \beta_{\text{LC}}$

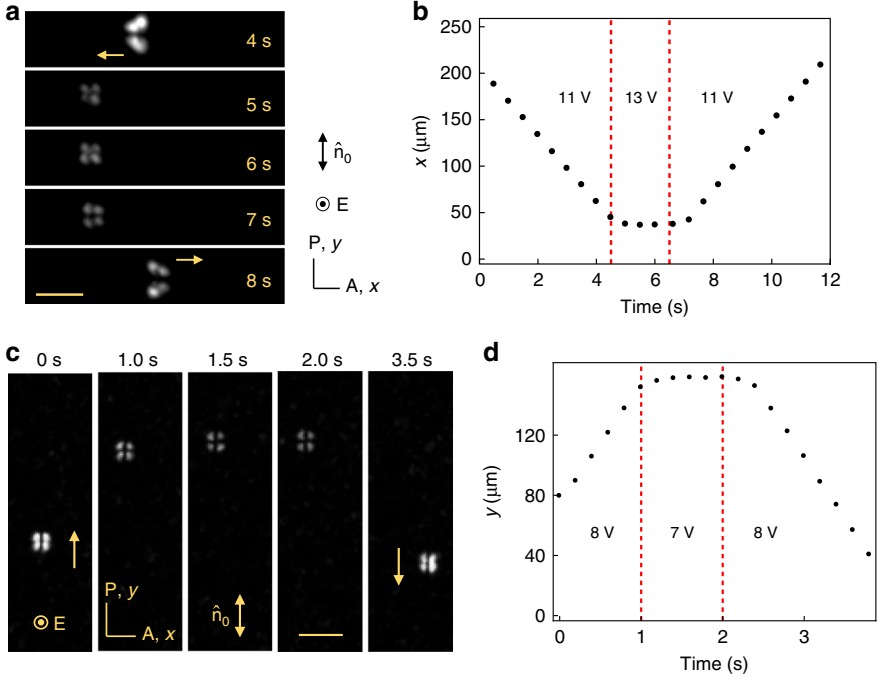

**Fig. 7 Velocity reversal of solitons by switching the voltage. a** Velocity reversal of $B_{90}^l$ soliton when the voltage is switched from 11.0 to 13.0 V and then back to 11.0 V. Scale bar 30 μm. **b** Dynamics of the $x$-location of the soliton $B_{90}^l$ shown in **a**. **c** Velocity reversal of $B_0^l$ soliton when the voltage is switched from 8.0 to 7.0 V and then back to 8.0 V. Scale bar 30 μm. **d** Dynamics of the $y$-location of the soliton $B_0^l$ shown in **c**

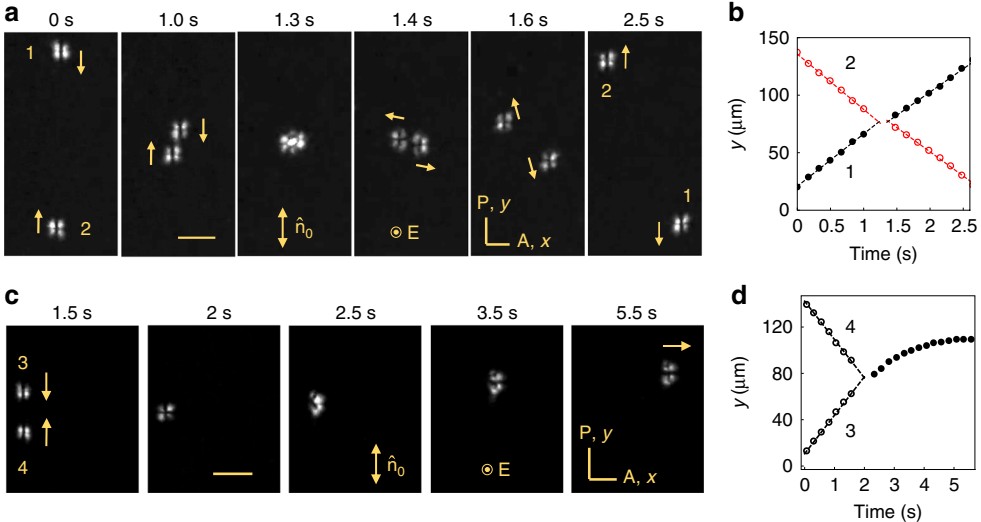

**Fig. 8 Collisions of $B_0^l$ solitons. a** Collision of two $B_0^l$ solitons ("1" and "2"); voltage 8.0 V, frequency 20 Hz, scale bar 30 μm. **b** Dynamics of the $y$-locations of the solitons shown in **a**. **c** Two $B_0^l$ solitons ("3" and "4") collide and transform into a single $B_{90}^l$ soliton. Voltage 8.5 V, frequency 20 Hz, scale bar 30 μm. **d** Dynamics of the $y$-locations of the solitons shown in **c**

diminishes the overall intensity of the transmitted light. Finally, numerical fitting using Eq. (1) allows us to estimate $|\theta_m| \approx 10$–$12°$.

## Discussion

The $B_\alpha^l$ solitons are sensitive to the anisotropy of dielectric permittivity, electric conductivity, concentration of ions, frequency, and amplitude of the field. For example, in CCN-47 kept at 55 °C, they exist in a narrow range of conductivities, $\sigma_\perp \approx (1.2$–$2.5) \times 10^{-8}\,\Omega^{-1}\,m^{-1}$. In this range, as $\sigma_\perp$ increases, the range of bullets stability shifts towards higher frequencies. Unlike the $(3+1)$D

bullets described previously for relatively high frequencies[13], $B_\alpha^l$ of the $(3+2)$D type exist at frequencies 27 Hz and lower. This region can be associated with the so-called "conductivity" regime limited from above by the critical frequency $f_c = \sqrt{\xi^2 - 1}/\tau_M$ introduced by Dubois-Violette et al.[19,20] to describe electrohydrodynamics of LCs. Here $\tau_M = \sigma_\perp/\varepsilon_0\varepsilon_\perp$ is the Maxwell relaxation time for a planar cell, $\varepsilon_0 = 8.85 \times 10^{-12}$ F/m, and $\xi^2 = \left(1 - \frac{\sigma_\perp}{\sigma_\parallel}\frac{\varepsilon_\parallel}{\varepsilon_\perp}\right)\left(1 + \frac{\alpha_2}{\eta_c}\frac{\varepsilon_\parallel}{\Delta\varepsilon}\right)$ is the material parameter that depends on conductivities ($\sigma_\parallel$ and $\sigma_\perp$), permittivities ($\varepsilon_\parallel$ and $\varepsilon_\perp$), and viscous coefficients ($\alpha_2$ and $\eta_c$). Using the experimental data measured at 5 kHz and 55 °C for

the TBABr-doped CCN-47, we find $1/\tau_M \sim 248$ Hz. The factor $\sqrt{\xi^2 - 1}$ in the expression for $f_c$ is hard to determine exactly, since $\alpha_2$ and $\eta_c$ are not known. Using the independently measured parameters of TBABr-doped CCN-47, we find $\left(1 - \frac{\sigma_\perp}{\sigma_\parallel}\frac{\varepsilon_\parallel}{\varepsilon_\perp}\right) \approx 0.3$ and $\frac{\varepsilon_\parallel}{\Delta\varepsilon} = -1.5$. With the ratio $\frac{-\alpha_2}{\eta_c}$ being on the order of 1, one finds that $\sqrt{\xi^2 - 1}$ is on the order of 1 or even smaller. This result implies that the dynamic solitons that appear at frequencies below 27 Hz correspond to the conductive regime of LC electro-hydrodynamics. Since the director oscillates with the same frequency as the field, the main reason of soliton existence is flexoelectric polarization and space charge developed at director deformations; both of these couple linearly to the field.

As illustrated by Figs. 1b and 2b, the time–average director asymmetry develops parallel to $\hat{\mathbf{n}}_0$ in $B_0^l$ and perpendicular to $\hat{\mathbf{n}}_0$ in $B_{90}^l$ case; the change in the field amplitude controls the type of asymmetry and thus the prevailing direction of propagation. Solitons that propagate under the angle $0 < \alpha < 90°$ show no mirror plane symmetry, Fig. 6a, b.

In summary, we demonstrate electrically driven multi-dimensional $(3 + 2)$D solitons–director bullets in a uniform nematic. These director bullets represent dissipative solitons that disappear once the driving electric field becomes too low or too high. By changing the amplitude of the driving electric field, the solitons can be controllably steered in the 2D plane perpendicular to the applied field; the background state remains homogeneous. The solitons represent perturbation of the director that are topologically trivial but self-confined along the longitudinal and transverse directions. In the $xy$ plane of propagation, the solitons width and length are much smaller than the corresponding dimension of the system. Along the $z$-axis, the stability of solitons is assisted by the surface anchoring at the bounding plates.

The azimuthal and polar tilts of the director oscillate with the same frequency as the driving AC electric field. The speed and propagation direction of the solitons is controlled by the field. When a soliton propagates along or perpendicularly to $\hat{\mathbf{n}}_0$, its dynamic director structure shows the mirror symmetry plane that contains the propagation direction. For intermediate propagation directions, $0 < \alpha < 90°$, the solitons show no mirror symmetry. The ability of the solitons to move along different directions opens a broad field of studies of their nontrivial interactions and collisions. It also promises practical applications. Controlling the in-plane dynamics of the solitons, one can develop devices for targeted 2D delivery of optical information. Furthermore, since the solitons represent director deformations and since director deformations attract colloidal particles[21], the $(3 + 2)$D solitons can be used as micro–cargo vehicles.

## Methods

**Materials.** We use a single-component nematic LC, 4′-butyl-4-heptyl-bicyclo-hexyl-4-carbonitrile, abbreviated as CCN-47 (Nematel GmbH). The conductivity was adjusted by adding 0.005 wt% TBABr, produced by Sigma-Aldrich, using chloroform as a solvent. Chloroform was evaporated in the vacuum oven for 24 h at the room temperature and then for another 24 h at 60 °C. We measured the anisotropy of both permittivity and conductivity by an LCR meter 4284A (Hewlett–Packard) using cells with planar and homeotropic alignment using polyimides AL1254 and JALS-204, respectively (both are purchased from Japan Synthetic Rubber Co.). The principal components of conductivity and dielectric permittivity tensors of doped CCN-47 are $\sigma_\parallel \approx 1.6 \times 10^{-8}\ \Omega^{-1}\,\text{m}^{-1}$, $\sigma_\perp \approx 1.8 \times 10^{-8}\ \Omega^{-1}\,\text{m}^{-1}$, $\varepsilon_\parallel \approx 4.9$, and $\varepsilon_\perp \approx 8.2$ at 5 kHz and 55 °C; the subscripts indicate whether the property is measured along the background director $\hat{\mathbf{n}}_0$ or perpendicularly to it.

**Generation of solitons.** The cell is composed of two glass substrates coated with indium tin oxide (ITO), which serve as the transparent electrode of active area $5 \times 5$ mm$^2$. The alignment layers PI-2555 coated on the surface of ITO were rubbed to provide a planar alignment. The temperature of the cell is controlled with a Linkam LTS350 hot stage and a Linkam TMS94 controller. The AC voltage is applied using

a waveform generator (Stanford Research Systems, Model DS345) and an amplifier (Krohn-hite Corporation, Model 7602).

**Optical characterization of solitons.** We use a polarizing Nikon TE2000 microscope equipped with two cameras: Emergent HR20000 with the maximum frame rate 1000 fps and MotionBLITZ EOSens mini1 (Mikrotron GmbH) with the maximum frame rate 6000 fps. The diagram in Fig. 4a was established by per-forming voltage scan with 0.1 V increments at $f =$ const; at each voltage level, the system was stabilized for 5 min before measurements. The location of soliton was tracked by an open-source software ImageJ and its plugin TrackMate. Measuring the $x,y$ coordinates of the solitons as a function of time yields the velocity. The azimuthal distortion of the director deviating from the background $\hat{\mathbf{n}}_0$ was determined by numerical simulations of the transmitted light intensity based on Jones matrix, in the geometry with two linear polarizers crossed at 78° and 65°. The nematic slab was split into thin layers, with the director twist $\varphi(i) = \varphi_m \sin\left(\frac{2i-1}{400}\pi\right)$, where $i$ is an integer in the range from 1 to 200 and $\varphi_m = \varphi_m(x, y, t)$ is the azimuthal distortion of the director in the middle of cell, $i = 100$. Using the measured birefringence of CCN-47 $\Delta n = 0.018$ at 55 °C, cell thickness $d = 8$ μm, wavelength of light $\lambda \approx 530$ nm, we calculate the ratio $T_{\text{bullet}}/T_0$ of the light intensity transmitted through the soliton to the light intensity transmitted through the background region, as a function of $\varphi_m$. Since $T_{\text{bullet}}/T_0$ equals the experimentally determined ratio $I_{\text{bullet}}/I_0$, where $I$ is the transmitted light intensity through the bullet and $I_0$ is the intensity of light transmitted through the background uniform region, the dependency $I_{\text{bullet}}/I_0$ on $\varphi_m$ allows us to map $\varphi_m(x, y, t)$ in Figs. 1b and 2b. The symmetry of director distortion of solitons is additionally verified by using complementary angles of polarizers' decrossing, such as 102° and 115°.

**Theory of light propagation through solitons.** We consider electromagnetic wave propagation in one-dimensionally distorted uniaxial nematic within Berreman's $4 \times 4$ matrix formalism[17] in the representation similar to refs. [22,23]. For a monochro-matic wave with the electric E and magnetic H fields changing as $\propto e^{-i\omega t}$, we use the dimensionless coordinates $\widetilde{\mathbf{r}} = \kappa\,\mathbf{r}$, where $\kappa = \omega/c = 2\pi/\lambda$ is the free space wave-number. We assume that the dielectric tensor at optical frequencies (optic tensor) $\varepsilon$ with the elements $\varepsilon_{ij}$ depends only on the coordinate $z$ perpendicular to the nematic slab, and does not depend on the in-plane coordinates $\boldsymbol{\rho} = \{\tilde{x}, \tilde{y}\}$; here and in what follows, tilde implies dimensionless coordinates, as specified above. The assumption preserves the in-plane 2D wave vector $\mathbf{q} = (q_1, q_2)$ and allows us to obtain the solution of the Maxwell equations in the form of four-vector $\bar{z}$-dependent ampli-tudes $\left(E_x, E_y, H_y, -H_x\right)\exp(i\mathbf{q}\rho)$, where $\mathbf{F} = (E_x, E_y, H_y, -H_x)$ obeys the equation:

$$\mathbf{F}' = i\,\mathbf{M}\cdot\mathbf{F}, \qquad (2)$$

where $' = \partial/\partial\tilde{z}$ and the $4 \times 4$ matrix M is:

$$\mathbf{M} = \begin{pmatrix} -q_1\varepsilon_{31}/\varepsilon_{33} & -q_1\varepsilon_{32}/\varepsilon_{33} & 1 - \left(q_1^2/\varepsilon_{33}\right) & -q_1 q_2/\varepsilon_{33} \\ -q_2\varepsilon_{31}/\varepsilon_{33} & -q_2\varepsilon_{32}/\varepsilon_{33} & -q_1 q_2/\varepsilon_{33} & 1 - \left(q_2^2/\varepsilon_{33}\right) \\ \varepsilon_{11} - q_1^2 - (\varepsilon_{13}\varepsilon_{31}/\varepsilon_{33}) & \varepsilon_{12} + q_1 q_2 - (\varepsilon_{13}\varepsilon_{32}/\varepsilon_{33}) & -q_1\varepsilon_{13}/\varepsilon_{33} & -q_2\varepsilon_{13}/\varepsilon_{33} \\ \varepsilon_{21} + q_1 q_2 - (\varepsilon_{23}\varepsilon_{31}/\varepsilon_{33}) & \varepsilon_{22} - q_2^2 - (\varepsilon_{23}\varepsilon_{32}/\varepsilon_{33}) & -q_1\varepsilon_{23}/\varepsilon_{33} & -q_2\varepsilon_{23}/\varepsilon_{33} \end{pmatrix} \qquad (3)$$

In a homogeneous medium, where $\boldsymbol{\varepsilon} = $ const and therefore $\mathbf{M} = \overline{\mathbf{M}} = $ const are $\tilde{z}$-independent, the general solution of Eq. (2) is

$$\overline{\mathbf{F}}(\tilde{z}) = \overline{\mathbf{V}}\cdot\overline{\mathbf{L}}(\tilde{z})\cdot\overline{\mathbf{A}}, \qquad (4)$$

where the matrices $\overline{\mathbf{V}} = \{\mathbf{V}_1, \mathbf{V}_2, \mathbf{V}_3, \mathbf{V}_4\}$ and $\bar{L}_{ij}(\tilde{z}) = \delta_{ij}\text{Exp}\left(i\bar{k}_i\tilde{z}\right)$ with $\alpha, \beta = 1, 2, 3, 4$ are built by the eigenvectors $\overline{\mathbf{V}}_i$ and the eigenvalues $\bar{k}_i$ of matrix $\overline{\mathbf{M}}$, that obey the equation

$$\overline{\mathbf{M}}\cdot\overline{\mathbf{V}}_i = \bar{k}_i\overline{\mathbf{V}}_i, \qquad (5)$$

and the four-vector $A$ contains the constant amplitudes of the eigenwaves that are determined by the boundary conditions.

When the uniaxial LC is distorted in the body of the soliton, $\hat{\mathbf{n}} = \hat{\mathbf{n}}(\tilde{z})$, the associated inhomogeneity of the optic tensor $\boldsymbol{\varepsilon}(\tilde{z}) = n_o^2\mathbf{I} + \left(n_e^2 - n_o^2\right)(\hat{\mathbf{n}}(\tilde{z})\otimes\hat{\mathbf{n}}(\tilde{z}))$ is controlled by the birefringence $\Delta n = n_e - n_o$, where $n_o$ and $n_e$ are the ordinary and extraordinary refractive indices, respectively. Because in our experiments the azimuthal distortions are small and obey the condition $|\varphi| < \Gamma^{-1}$, where $\Gamma = 2\pi\Delta nd/\lambda \approx 1.7$ is the phase retardation of the undistorted LC area, we use the perturbation method to determine the transformation between ordinary and extraordinary waves.

We split $\mathbf{M}(\tilde{z}) = \overline{\mathbf{M}} + \widetilde{\mathbf{M}}(\tilde{z})$ into a homogeneous part $\overline{\mathbf{M}}$ and an inhomogeneous $\widetilde{\mathbf{M}}(\tilde{z})$ part. The solution of Eq. (2) for this case can be presented as:

$$\mathbf{F}(\tilde{z}) = \overline{\mathbf{V}}\cdot\overline{\mathbf{L}}(\tilde{z})\cdot\mathbf{A}(\tilde{z}), \qquad (6)$$

where the matrices $\overline{\mathbf{V}}$ and $\bar{L}_{ij}(\tilde{z}) = \delta_{ij}\text{Exp}\left(i\bar{k}_i\tilde{z}\right)$ correspond to the homogeneous part and obey Eq. (5). Substituting this solution into Eq. (2), we obtain the equation that defines evolution of the eigenwaves' amplitudes four-vector $\mathbf{A}(\tilde{z})$ along the $z$-axis

$$\mathbf{A}'(\tilde{z}) = \overline{\mathbf{L}}^{-1}(\tilde{z})\cdot\widetilde{\mathbf{S}}(\tilde{z})\cdot\overline{\mathbf{L}}(\tilde{z})\cdot\mathbf{A}(\tilde{z}), \qquad (7)$$

where $\bar{\mathbf{S}}(\tilde{z}) = i\,\bar{\mathbf{V}}^{-1} \cdot \tilde{\mathbf{M}}(\tilde{z}) \cdot \bar{\mathbf{V}}$ is the matrix, which changes the amplitudes of eigenwaves of $\bar{\mathbf{M}}$. The linearity of Eq. (7) allows us to separate boundary conditions from evolution of the amplitudes of eigenmodes in the bulk of the inhomogeneous medium, by introducing the propagation matrix $\mathbf{U}(\tilde{z})$,

$$\mathbf{A}(\tilde{z}) = \mathbf{U}(\tilde{z}) \cdot \mathbf{A}(0). \qquad (8)$$

The propagation matrix $\mathbf{U}(\tilde{z})$ obeys the boundary condition $U(0) = I_4$, where $I_4$ is the $4 \times 4$ unit matrix and the equation similar to Eq. (7),

$$\mathbf{U}'(\tilde{z}) = \bar{\mathbf{L}}^{-1}(\tilde{z}) \cdot \widetilde{\mathbf{S}}(\tilde{z}) \cdot \bar{\mathbf{L}}(\tilde{z}) \cdot \mathbf{U}(\tilde{z}). \qquad (9)$$

The solution of the last equation in the approximation linear on $\widetilde{\mathbf{S}}(\tilde{z})$,

$$U_{ij}(\tilde{z}) = \delta_{ij} + \int_0^{\tilde{z}} \widetilde{S}_{ij}(\bar{z}) \mathrm{Exp}\left[i\left(\bar{k}_j - \bar{k}_i\right)\bar{z}\right] d\bar{z}, \qquad (10)$$

is valid if the integral contributions are smaller than 1.

Because of the strong planar anchoring at the bounding plates, $\hat{\mathbf{n}}(0) = \hat{\mathbf{n}}(\tilde{d}) = \hat{\mathbf{n}}_0 = (0, 1, 0)$, we select $\bar{\mathbf{M}}$ that corresponds to $\hat{\mathbf{n}}_0$ and assume that the polar angle $\theta(\tilde{z}) = \theta_m \sin(\pi\tilde{z}/\tilde{d})$ and the azimuthal angle $\varphi(\tilde{z}) = \varphi_m \sin(\pi\tilde{z}/\tilde{d})$, which define the director field $\hat{\mathbf{n}} = (\cos\theta\sin\varphi, \cos\theta\cos\varphi, \sin\theta)$ have simple one harmonic $z$-dependence with small amplitudes $\theta_m$ and $\varphi_m$. The transmitted light intensity in the optical scheme with two crossed linear polarizers is determined by $\left|U_{12}\left(\tilde{d}\right)\right|^2$.

When the $xz$ is the incidence plane, $\left|U_{12}\left(\tilde{d}\right)\right|^2 = \frac{4\Gamma^2}{\pi^2}\left[\varphi_m - \theta_m\tan\beta_{LC}\right]^2 G(\Gamma)$, where $G(\Gamma) \approx 1 - 0.04\Gamma^2$ is a slowly decaying function and $\beta_{LC}$ is the angle of light propagation in the LC medium, measured with respect to the $z$-axis. When $yz$ is the incidence plane, then $|U_{12}(\Gamma)|^2 = \frac{2\Gamma^2}{\pi}\varphi_m^2\left[1 + \left(\frac{\pi}{4}\theta_m\tan\beta_{LC}\right)^2\right]G(\Gamma)$. Taking into account the light leakage while the beam propagates between the crossed polarizers, we result in the transmitted light intensity $I$, normalized by the incident intensity, aszers, we result in the transmitted light intensity $I$, normalized by the incident intensity, as

$$I \approx I_{\mathrm{leak}} + \left|U_{12}\left(\tilde{d}\right)\right|^2, \qquad (11)$$

where $I_{\mathrm{leak}}$ is the normalized intensity of the leaked light.

## Data availability

The data that support the findings of the study are available from the corresponding author upon reasonable request.

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

## Acknowledgements

The work was supported by NSF grants DMS-1729509 (experiments and analysis) and DMR-1905053 (analysis and presentation). Useful discussions with M.C. Calderer and D. Golovaty are appreciated.

## Author contributions

B.L. discovered the studied solitons. S.V.S. developed the model of light transmittance through the soliton. B.L. and R.X. performed the experimental studies. S.P. measured the conductivity of the studied materials. B.L., R.X., S.V.S., and O.D.L. analyzed the data and discussed the results. O.D.L. directed the research and wrote the manuscript with an input from all coauthors.

## Additional information

**Competing interests:** The authors declare no competing interests.

