## [Peer Review File · Nature Communications]

Reviewers' comments:

Reviewer #1 (Remarks to the Author):

The paper reports on the realization and detailed experimental characterization of localized waves (called director bullets) in a nematic liquid crystal cell, suitably designed and fabricated to obtain the onset of the observed phenomena. The investigated propagating structures are solitons of (3+2)D type; interestingly, with respect to the background director orientation, these director bullets show different kinds of asymmetry, which can be controlled by applying an external electric field. The argument is of present scientific interest and all reported results, which are definitely new, are exhaustively illustrated and explained with an adequate number of figures and movies. The English is good and the paper is of straightforward reading notwithstanding its scientific accuracy. In my opinion, in order to deserve publication, it is enough that the authors address the following comments.

- 1) In the introduction, it would be appropriate to give an (even very brief) outline of the theoretical model exposed in the supplementary material, trying to frame the large number of performed experiments and measurements in a "design of investigation" of the observed phenomenon. This would avoid the risk that the work is seen only as a "collection of observed effects", very nice but not clearly related to each other, or whose link has not been identified.
- 2) In the same spirit of comment 1, just a sentence taken from the theoretical model could be used to explain the values of the used frequency range, reported in line 58 of page 3 ($5\text{Hz} \leq f \leq 27\text{Hz}$), which, otherwise, at a first glance appear to the reader as "magic numbers" of obscure origin.
- 3) The choice of the range values of the used voltage ($4\text{V} \leq U \leq 12\text{V}$, line 58 of page 3) has an unclear relation with the model and should be explained.
- 4) The choice of the two values for the angle between the polarizers (78° and 65° , line 101 of page 5) has an unclear origin and should be explained.
- 5) The choice of the two values for the incidence angle (0° and 15° , line 126 of page 8) has an unclear origin and should be explained.
- 6) There is a misprint in the title of the supplementary material.

Reviewer #2 (Remarks to the Author):

The manuscript reports a study onto the three-dimensional solitary waves with the electrically tunable direction of propagation in nematics liquid crystal. More precisely, the authors show an experimental realization of multidimensional solitary waves of a $(3+2)D$ type which are steered by the electric field in a plane perpendicular to the field. In the present form, I do not recommend the manuscript for publication. The results found are interesting and appealing. However, interpretation and description are incorrect. Below I present the arguments that lead me to my conclusion

*.- The authors have a profound confusion of concepts. Localized waves and solitons are dynamic behaviors of conservative systems. Liquid crystals are dissipative systems. There is a long literature of the counterpart of dissipative solitons, usually called localized structures, see for example

- i) Dissipative Solitons: From Optics to Biology and Medicine, Lecture Notes in Physics Vol. 751, edited by N. Akhmediev and A. Ankiewicz (Springer, Heidelberg, 2008).
- ii) Ackemann T, Firth WJ, Oppo GL. 2009 Fundamentals and applications of spatial dissipative solitons in photonic devices. In Advances in atomic, molecular, and optical physics, vol. 57 (eds E Arimondo, PR Berman, CC Lin), pp. 323–421. Burlington, VT: Academic Press.
- iii) H. G. Purwins, H. U. Bodeker, and S. Amiranashvili, “Dissipative solitons,” Adv. Phys. 59, 485 (2010).
- iv) Descalzi, M. G. Clerc, S. Residori, and G. Assanto, Localized States in Physics: Solitons and Patterns (Springer, 2011).
- v) E. Knobloch, Spatial localization in dissipative systems, Annu. Rev. Condens. Matter Phys. 6, 325 (2015).

The localized states observed by the authors are localized structures and not solitary waves.

*.- It is important to note that nematicons $(2 + 1)D$ have only been observed in thin films of liquid crystals or liquid crystal cells, that is, a dimension is constricted. Then, the nematicons are effective objects of $(1 + 1)D$. All the theoretical description is in $1 + 1$ dimensions. This discussion must be clear in the manuscript.

*.- Likewise, the results presented by the authors is in a liquid crystal cell, a dimension is strongly constrained. Note that the original solitons of Russell, are three-dimensional objects are waves in a channel, however, its description and nature is $(1 + 1) D$. Hence, the results presented by the authors

correspond to localized structures $(2 + 1)D$. The direction of propagation is an effect of initial conditions and external factors such as electric fields.

*.- The notation $(n + m) D$ is confusing, because instantaneously the localized structures propagate in a single direction, that is, $(n + 1) D$.

*.- Experimentally, it has been reported a supercritical transition from stationary to moving localized structures $[(3+2)D]$ in a planar gas-discharge system H.U. Bodeker, M.C. Rottger, A. W. Liehr, T.D. Frank, R. Friedrich, and

H. G. Purwins, Phys. Rev. E 67, 056220 (2003). Hence, the propagative localized structures $[(3+2)D]$ are well known in the literature.

*.- Localized asymmetric propagative spots were theoretically envisaged in reference Alvarez-Socorro, Clerc, and M. Tlidi CHAOS 28, 053119 (2018). A clarifying comment is necessary.

Minor comment

*.- To facilitate the concentration of the reader it is recommended to leave the details of the liquid crystal in the methods section.

Dear Dr. Dubrovina,

We thank you and reviewers for evaluation of our work. We are thankful for the valuable comments that help us to improve the presentation. Below we list all the comments and our replies. We take advantage of the Nature Communications opportunities to significantly expand the text and illustrative material as compared to our original submission to Nature.

Reviewer #1 (Remarks to the Author):

Q1.1 The paper reports on the realization and detailed experimental characterization of localized waves (called director bullets) in a nematic liquid crystal cell, suitably designed and fabricated to obtain the onset of the observed phenomena. The investigated propagating structures are solitons of (3+2)D type; interestingly, with respect to the background director orientation, these director bullets show different kinds of asymmetry, which can be controlled by applying an external electric field. The argument is of present scientific interest and all reported results, which are definitely new, are exhaustively illustrated and explained with an adequate number of figures and movies. The English is good and the paper is of straightforward reading notwithstanding its scientific accuracy. In my opinion, in order to deserve publication, it is enough that the authors address the following comments.

A1.1 We thank Reviewer #1 for stating that “*The argument is of present scientific interest and all reported results, which are definitely new, are exhaustively illustrated and explained with an adequate number of figures and movies. The English is good, and the paper is of straightforward reading notwithstanding its scientific accuracy*”.

Q1.2 In the introduction, it would be appropriate to give an (even very brief) outline of the theoretical model exposed in the supplementary material, trying to frame the large number of performed experiments and measurements in a "design of investigation" of the observed phenomenon. This would avoid the risk that the work is seen only as a "collection of observed effects", very nice but not clearly related to each other, or whose link has not been identified.

A1.2 We thank the reviewer for the suggestion and the opportunity to expand the text. We add the following text in the introduction (pages 3-4).

“**The dynamic director structure of the solitons is established by polarizing optical microscopy in transmission mode with ultrafast videocamera. The patterns of transmitted light intensity for normal and oblique incidence are analyzed by the theory based on Berreman’s 4 x 4 matrix formalism¹⁷. The analysis establishes that the director deformations within the solitons oscillate with the frequency of the driving electric field. This feature points to the linear coupling of $\hat{n}(\mathbf{r})$ to the electric field as a main reason for the bullets stability.**”

Furthermore, to help the readers, we introduced subsections’ titles in the manuscript that explain clearly which feature is being discussed.

Q1.3 In the same spirit of comment 1, just a sentence taken from the theoretical model could be

used to explain the values of the used frequency range, reported in line 58 of page 3 ($5\text{Hz} \leq f \leq 27\text{ Hz}$), which, otherwise, at a first glance appear to the reader as "magic numbers" of obscure origin.

A1.3 We thank the reviewer for the suggestion. We added the text below to explain what we currently know about the frequency range in which the solitons are stable (page 8).

“The experimentally established range underlines that the B_α^1 stability results from a fine balance of different mechanisms of the electric field-nematic coupling, such as flexoelectric polarization, anisotropy of dielectric permittivity, conductivity and its anisotropy. Absence of solitons at very low frequencies $f < 5\text{ Hz}$ and high voltages $U > 13\text{ V}$ is apparently caused by screening of electrodes by ions.”.

Q1.4 The choice of the range values of the used voltage ($4\text{V} \leq U \leq 12\text{V}$, line 58 of page 3) has an unclear relation with the model and should be explained.

A1.4 We thank the reviewer for the suggestion. We added the text above in A1.3 to explain what we currently know about the voltage range in which the solitons are stable (page 8).

Q1.5 The choice of the two values for the angle between the polarizers (78° and 65° , line 101 of page 5) has an unclear origin and should be explained.

A1.5 We explained the origin of our choice in the text (pages 14-15) as follows:

“Usually, polarizing microscopy is performed with the two polarizers crossed at 90° . However, such standard approach does not allow one to distinguish the states φ and $-\varphi$. The decrossed polarizers allow one to distinguish φ and $-\varphi$; the angular values of decrossing, 78° or 65° , have been found experimentally to yield a good contrast between the orientations φ and $-\varphi$.”.

Q1.6 The choice of the two values for the incidence angle (0° and 15° , line 126 of page 8) has an unclear origin and should be explained.

A1.6 We explained the choice in the text (pages 15-16), by adding the results of theoretical analysis, as follows:

“To prove these polar oscillations, we measured the dynamics of light transmittance through the cell and two crossed polarizers at oblique incidence of the probing light beam, $\beta = 15^\circ$, Fig.2e, where β is the angle between the probing beam and the normal to the cell measured outside the cell. The theoretical model in Methods, Equation (11), predicts that for B_{90}^1 solitons, when the incidence plane xz is perpendicular to \hat{n}_0 , the transmitted light intensity I , normalized by the incident intensity, depends on the angle of incidence β , the azimuthal φ_m and polar θ_m director angles measured in the middle plane,

$$I \propto I_{\text{leak}} + \frac{4\Gamma^2}{\pi^2} (1 - 0.04\Gamma^2) (\varphi_m - \theta_m \tan \beta_{LC})^2, \quad (1)$$

where I_{leak} is the normalized light intensity leaked while the beam propagates between the crossed polarizers and the cell, $\Gamma = 2\pi\Delta nd/\lambda \approx 1.7$ is the phase retardation of the undistorted nematic, determined by the wavelength $\lambda = 530$ nm and birefringence $\Delta n = n_e - n_o$, n_o and n_e are the ordinary and extraordinary refractive indices, respectively; $\beta_{\text{LC}} = \beta/n_o$ is the angle at which the probing beam propagates inside the nematic. For $\beta_{\text{LC}} = \beta = 0$, I depends only on φ_m ; however, for $\beta_{\text{LC}} = \beta \neq 0$, Equation (1) demonstrates that the measured light intensity depends on the sign of θ_m , which allows us to determine the frequency of its oscillations. We found experimentally that the value $\beta = 15^\circ$ (which corresponds to $\beta_{\text{LC}} = \beta/n_o = 10.6^\circ$) provides the best contrast between θ_m and $-\theta_m$. The experiment, Fig. 2e, demonstrates that the transmission peaks for the two half-periods of the applied field are different for both normal $\beta_{\text{LC}} = \beta = 0$ and oblique $\beta = 15^\circ$ light incidence. Since $I \propto \varphi_m^2$, Equation (1), the asymmetry of the two half-periods indicates that angle φ_m has a nonzero average value and changes with the frequency f . The experiment and modeling for the oblique $\beta = 15^\circ$ light incidence, Fig.2e, show that θ_m oscillates with the same frequency f . “

We also move the entire section on theory from Supplement to Methods.

Q1.7 There is a misprint in the title of the supplementary material.

A1.7 We deleted the misprint “of” in the title.

Reviewer #2 (Remarks to the Author):

Q2.1 The manuscript reports a study onto the three-dimensional solitary waves with the electrically tunable direction of propagation in nematics liquids crystal. More precisely, the authors show an experimental realization of multidimensional solitary waves of a (3+2)D type which are steered by the electric field in a plane perpendicular to the field. In the present form, I do not recommend the manuscript for publication. The results found are interesting and appealing. However, interpretation and description are incorrect. Below I present the arguments that lead me to my conclusion

A2.1 We thank Reviewer #2 for stressing that the results are interesting and appealing. We strongly believe that the interpretation and description are correct, as we present a careful experimental description of novel structures and give an interpretation of their substance, which is rooted in the behavior of the director field. It is true that we did not discuss enough the nomenclature in the field, mainly because our original submission was to Nature, with a strict length restriction. Now that we transfer to Nature Communications, we could describe in a greater detail the place that our structures occupy in the general schematics of solitons and discuss the relevant nomenclature, as described below and in the modified text.

Q2.2 The authors have a profound confusion of concepts. Localized waves and solitons are dynamic behaviors of conservative systems. Liquid crystals are dissipative systems. There is a long literature of the counterpart of dissipative solitons, usually called localized structures, see for example,

i) Dissipative Solitons: From Optics to Biology and Medicine, Lecture Notes in Physics Vol. 751, edited by N. Akhmediev and A. Ankiewicz (Springer, Heidelberg, 2008).

ii) Ackemann T, Firth WJ, Oppo GL. 2009 Fundamentals and applications of spatial dissipative solitons in photonic devices. In Advances in atomic, molecular, and optical physics, vol. 57 (eds E Arimondo, PR Berman, CC Lin), pp. 323–421. Burlington, VT: Academic Press.

iii) H. G. Purwins, H. U. Bodeker, and S. Amiranashvili, “Dissipative solitons,” Adv. Phys. 59, 485 (2010).

iv) Descalzi, M. G. Clerc, S. Residori, and G. Assanto, Localized States in Physics: Solitons and Patterns (Springer, 2011).

v) E. Knobloch, Spatial localization in dissipative systems, Annu. Rev. Condens. Matter Phys. 6, 325 (2015).

The localized states observed by the authors are localized structures and not solitary waves.

A2.2 We thank the reviewer for the provided list of references, all of which are included in the revised manuscript. We stressed repeatedly in our manuscript that the structures exist only because there is an applied field, which is precisely related to the dissipative character of the system. The concept of a “dissipative soliton” is a subset of the broad “soliton” concept, as clear from the current literature; eg the abstract of the book “Dissipative Solitons: From Optics to Biology and Medicine” states that “*The dissipative soliton concept is a fundamental extension of the concept of solitons in conservative and integrable systems. It includes ideas from three major sources, namely standard soliton theory developed since the 1960s, nonlinear dynamics theory, and Prigogine’s ideas of systems far from equilibrium.*” Since our structures are experimentally proven to be waves of the director and proven to behave as particles in collisions, the terms “soliton” and “solitary waves” are correct. Of course, these structures can also be called “localized structures” and “dissipative solitons”, which we do in the revised manuscript. We respectfully draw the attention of the Reviewers and editors to the fact that in the paper “Spatially localized structures in dissipative systems: Open problems”, Nonlinearity **21** T45 (2008), a leading expert on dissipative solitons (recommended by Referee #2), E. Knobloch, states on page T54 regarding “forced symmetry-breaking of localized states” that “The steady states in general turn into drifting or travelling pulses that we may call solitary waves.” (highlighted by us). Since our structures are mobile and engage in collisions as particles, this statement overturns the criticism of Reviewer#2 that “the localized structures observed by the authors are localized structures and not solitary waves”. We emphasized that the director bullets in nematics are dissipative solitons, by stating in the introduction (page 2)

“Among the broad family of solitons there is a class of localized structures, often called dissipative solitons⁷⁻¹¹. Dissipative solitons require an external driving; they represent a portion of a pattern surrounded by a homogeneous steady state; below a certain strength of the driver,

they vanish⁷⁻¹¹. Experimentally, dissipative solitons were realized in the form of electric current filaments in a 2D planar gas-discharge system¹² and also as (3+1)D particle-like electrically powered solitary waves of molecular orientation (specified by the so-called director $\hat{\mathbf{n}}$) in a nematic liquid crystal¹³. These (3+1)D solitons, called “director bullets”, represent spatially-confined perturbations of the director field $\hat{\mathbf{n}}(\mathbf{r})$ that coexist with a uniform director state $\hat{\mathbf{n}}_0 = \text{const}$. The distortions $\hat{\mathbf{n}}(\mathbf{r})$ oscillate with the same frequency as the frequency f of the driving electric field. They disappear when the amplitude of the electric field becomes too low and when $f < 100$ Hz.”

We also stated in the Concluding part, page 18, “These director bullets represent dissipative solitons that disappear once the driving electric field becomes too low or too high.”

Q2.3 It is important to note that nematicons (2 + 1)D have only been observed in thin films of liquid crystals or liquid crystal cells, that is, a dimension is constricted. Then, the nematicons are effective objects of (1+1)D. All the theoretical description is in 1 + 1 dimensions. This discussion must be clear in the manuscript.

A2.3 Since the dimensionality of nematicons is not the subject of our paper, to comply with the Reviewer#2 remark, we modified the text by eliminating the dimensionality statement, on page 3, line 58: “One example are nematicons, self-focused light beams propagating in a nematic.”

Although we made this modification, we would like to note that the nematicons are (2+1)D formations, as the director corridor and the laser beam intensity are spatially varying along the normal to the cell and in the plane of the cell. Please note that the spatial optical solitons were observed in liquid crystals also in other geometries (a cylindrical capillary), in the paper entitled “A thermal (2D+1) spatial optical soliton in a dye doped liquid crystal” by Derrien et al, J. Opt. A: Pure Appl. Opt **2**, 332 (2000) (highlights by us), where both the experiment and the theory are demonstrating (2+1)D nature, as the title of the paper clearly stresses. Similarly, in the paper by Conti et al “Observations of optical spatial solitons in a highly nonlocal medium” Phys Rev Lett **92**, 113902 (2004), the experiment and theory show a (2+1)D nature of nematicons, see also the review paper by Assanto, Physics Reports **516**, 147-208 (2012).

Q2.4 Likewise, the results presented by the authors is in a liquid crystal cell, a dimension is strongly constrained. Note that the original solitons of Russell, are three-dimensional objects are waves in a channel, however, its description and nature is (1 + 1) D. Hence, the results presented by the authors correspond to localized structures (2 + 1)D. The direction of propagation is an effect of initial conditions and external factors such as electric fields.

A2.4 We respectfully disagree. Our solitary waves are similar to the (3+1)D director bullets described previously by our group, Ref.13. The difference is in the possibility to steer the bullets in the 2D plane perpendicular to the electric field. The newly observed bullets propagate in the deterministic fashion either parallel to the director, or perpendicular to it, or, when the electric field changes, under a tilted trajectory. These structures are very different from the localized

structures in isotropic systems observed in *H.U. Bodeker, M.C. Rottger, A. W. Liehr, T.D. Frank, R. Friedrich, and H. G. Purwins, Phys. Rev. E 67, 056220 (2003)* and *Alvarez-Socorro, A. J., Clerc, M. G. & Tlidi, M. Spontaneous motion of localized structures induced by parity symmetry breaking transition. Chaos: An Interdisciplinary Journal of Nonlinear Science 28, 053119, (2018)*. In these publications, the authors stressed that their dissipative solitons/localized structures behave as Brownian particles (passive or active) with random propagations, since their system is isotropic. Our system is anisotropic and the direction of propagation is related to the in-plane director alignment, as we stress in the manuscript. Note also that the field is applied normally to the cell, thus it is not the field alone that determines the direction of propagation. To explain the features, we added the following text on page 3:

“In this work, we demonstrate experimentally multidimensional steerable solitary waves of a (3+2)D director bullets type. These director bullets represent waves of director deformations driven by an alternating current (AC) electric field. The solitons form in a planar homogeneous sandwich-like nematic cell, with the ground-state director $\hat{\mathbf{n}}_0$ being parallel to the bounding plates and to planar transparent electrodes. The electric field, applied along the normal to the cell, is perpendicular to $\hat{\mathbf{n}}_0$. The waves are self-localized in three spatial dimensions, while the background $\hat{\mathbf{n}}_0$ remains unperturbed.”

And on page 4:

“Most interestingly, depending on the electric field amplitude, the solitons show different geometries of symmetry breaking of the director deformations $\hat{\mathbf{n}}(\mathbf{r})$ and because of that, they can move either parallel to $\hat{\mathbf{n}}_0$ or perpendicular to it. The electric field, being perpendicular to the bullets trajectories, provides no sense of direction by itself; it is the in-plane anisotropy of the system and its coupling to the field that produces different scenarios of symmetry breaking and allows one to steer the solitons by changing the electric field. To stress the 2D control over trajectories of the director bullets, we classify them as (3+2)D solitons. To the best of our knowledge, there are no other examples of stable and steerable multidimensional solitons that move through a homogeneous background along externally controlled trajectories, neither among the “classic” solitons in conservative systems (such as the Russell’s solitary wave) nor among the localized structures/dissipative solitons in driven dissipative systems.”

And on page 9-10:

“Deterministic directional propagation of B_α^l bullets differs dramatically from the stochastic motion of localized structures of current filaments described experimentally¹² and theoretically¹⁴. The obvious reason is the anisotropy of the nematic cell. Similarly to the (3+1)D B_{90}^h solitons described previously¹³, propagation direction of the B_α^l bullets is associated with the internal symmetry breaking. However, the important new feature of B_α^l is that this internal symmetry breaking can be either fore-aft (B_0^l , Fig.1) or left-right (B_{90}^l , Fig.2), or shows no mirror symmetry at all, Fig.4.”

and page 14:

“This intrinsic symmetry breaking of the director field is reminiscent of the symmetry breaking that leads to a transition from the stationary to moving localized structures/dissipative solitons described by Alvarez-Socorro et al¹⁴ with that difference that the nematic background is anisotropic and thus the spectrum of possible symmetry breaking scenarios for director bullets is much broader, ranging from left-right to fore-aft mirror symmetries and to structures that have no mirror symmetry at all.”

Q2.5 The notation $(n + m) D$ is confusing, because instantaneously the localized structures propagate in a single direction, that is, $(n + 1) D$.

A2.5 We show that our solitons’ trajectories can explore deterministically the entire 2D plane, see Fig.4. Since we prove the 2D trajectories experimentally, we preserve the nomenclature to describe properly our findings.

We added the explanation on page 4: “Most interestingly, depending on the electric field amplitude, the solitons show different geometries of symmetry breaking of the director deformations $\hat{\mathbf{n}}(\mathbf{r})$ and because of that, they can move either parallel to $\hat{\mathbf{n}}_0$ or perpendicular to it. The electric field, being perpendicular to the bullets trajectories, provides no sense of direction by itself; it is the in-plane anisotropy of the system and its coupling to the field that produces different scenarios of symmetry breaking and allows one to steer the solitons by changing the electric field. To stress the 2D control over trajectories of the director bullets, we classify them as $(3+2)D$ solitons”.

Q2.6 Experimentally, it has been reported a supercritical transition from stationary to moving localized structures $[(3+2)D]$ in a planar gas-discharge system H.U. Bodeker, M.C. Rottger, A. W. Liehr, T.D. Frank, R. Friedrich, and H. G. Purwins, Phys. Rev. E 67, 056220 (2003). Hence, the propagative localized structures $[(3+2)D]$ are well known in the literature.

A2.6 We cite and discuss the recommended references in the revised manuscript, page 2:

“Experimentally, dissipative solitons were realized in the form of electric current filaments in a 2D planar gas-discharge system¹²,”

The previously reported structures are filaments that connect the two plate electrodes, thus these structures are not 3D but 2D, as stressed in the paper Alvarez-Socorro, A. J., Clerc, M. G. & Tlidi, M. *Spontaneous motion of localized structures induced by parity symmetry breaking transition. Chaos: An Interdisciplinary Journal of Nonlinear Science* 28, 053119, (2018)], page 2 “Experimental observation of a supercritical transition from stationary to moving localized structures has been realized in two-dimensional planar gas-discharge systems [H.U. Bodeker, M.C. Rottger, A. W. Liehr, T.D. Frank, R. Friedrich, and H. G. Purwins, Phys. Rev. E 67, 056220 (2003)] and on page 5: “Most of the experimental observations of localized structures

have been realized in two dimensional systems, in which stationary localized structures are observed. Experimentally, it has been reported a supercritical transition from stationary to moving localized structures on a **planar gas-discharge** system” (highlighted by us).

Furthermore, these filaments described in PRE (2003) show a random Brownian motion caused by noise, thus they cannot be classified as deterministically exploring the 2D plane. It is stressed in the paper *Alvarez-Socorro, A. J., Clerc, M. G. & Tlidi, M. Spontaneous motion of localized structures induced by parity symmetry breaking transition. Chaos: An Interdisciplinary Journal of Nonlinear Science 28, 053119, (2018)*: “The direction in which localized structures propagate depends on the initial conditions used. **Indeed, there is no preferred direction since the system is isotropic.**” (highlights by us). This is a crucial difference with our system that is **anisotropic and provides clear preferred directions of soliton propagation.** We clarify the issue on pages 9-10:

Deterministic directional propagation of B_α^l bullets differs dramatically from the stochastic motion of localized structures of current filaments described experimentally¹² and theoretically¹⁴. The obvious reason is the anisotropy of the nematic cell. Similarly to the (3+1)D B_{90}^h solitons described previously¹³, propagation direction of the B_α^l bullets is associated with the internal symmetry breaking. However, the important new feature of B_α^l is that this internal symmetry breaking can be either fore-aft (B_0^l , Fig.1) or left-right (B_{90}^l , Fig.2), or shows no mirror symmetry at all, Fig.4.

Q2.7 Localized asymmetric propagative spots were theoretically envisaged in reference Alvarez-Socorro, Clerc, and M. Tlidi CHAOS 28, 053119 (2018). A clarifying comment is necessary.

A2.7 We thank for the reference and included it in the revised manuscript, commenting as follows:

On page 3:

“A similar case of localized structures moving in one spatial dimension as a result of an internal symmetry breaking instability has been described independently and simultaneously by Alvarez-Socorro, Clerc and Tlidi¹⁴ for a 2D isotropic system. In contrast to the case of (3+1)D director bullets, the motion direction of localized structures considered in Ref. 14 is arbitrary (defined by the initial condition used in simulations): a preferred direction does not exist since the system is isotropic¹⁴.”

and on pages 9-10:

“Deterministic directional propagation of B_α^l bullets differs dramatically from the stochastic motion of localized structures of current filaments described experimentally¹² and theoretically¹⁴. The obvious reason is the anisotropy of the nematic cell. Similarly to the (3+1)D B_{90}^h solitons described previously¹³, propagation direction of the B_α^l bullets is associated with the internal symmetry breaking. However, the important new feature of B_α^l is that this internal

symmetry breaking can be either fore-aft (B_0^l , Fig.1) or left-right (B_{90}^l , Fig.2), or shows no mirror symmetry at all, Fig.4.”

and on page 14:

“This intrinsic symmetry breaking of the director field is reminiscent of the symmetry breaking that leads to a transition from the stationary to moving localized structures/dissipative solitons described by Alvarez-Socorro et al ¹⁴ with that difference that the nematic background is anisotropic and thus the spectrum of possible symmetry breaking scenarios for director bullets is much broader, ranging from left-right to fore-aft mirror symmetries and to structures that have no mirror symmetry at all.”

Q.2.8 To facilitate the concentration of the reader it is recommended to leave the details of the liquid crystal in the methods section.

A2.8 We introduced subsections’ titles which would help the readers to follow the presentation. We also moved the theory of light propagation from Supplement to Methods, as requested by Reviewer#1.

We hope that our modifications clarify the issues raised by the reviewers to whom we are thankful for careful evaluation. We hope that the revised manuscript is suitable for publication in Nature Communications.

On behalf of all authors,
Oleg D. Lavrentovich

AUTHORS REPLIES TO REVIEWERS' COMMENTS

Reviewer #1 (Remarks to the Author):

R1: The authors have exhaustively addressed in details all my comments and the paper is now suitable for publication.

Cesare Umeton

A1. We thank Reviewer 1 for the evaluation of our work and for the recommendation to publish.

Reviewer #2 (Remarks to the Author):

R2: In the revised version, the authors have adequately answered most of the observations raised, in the previous report. The revised version has been extended to clarify some points and be more accessible to a broad community.

Detail of the reason for choosing specific experimental parameters has been incorporated. In the revised version, the basic concepts under the study have been clarified, which, in my opinion, now allows a correct interpretation of the results found. Besides, Authors have included in the manuscript all suggestions. The quality and accessibility of the manuscript have been improved. Hence, I recommend the publication of this manuscript in Nature Communications.

A2 We thank Reviewer 2 for the evaluation of our work and for the recommendation to publish.